# Advancements in Biochar Modification for Enhanced Phosphorus Utilization in Agriculture

Nazir Ahmed [1,†], Lifang Deng [2,†], Chuan Wang [1], Zia-ul-Hassan Shah [3], Lansheng Deng [4], Yongquan Li [1], Juan Li [1], Sadaruddin Chachar [1], Zaid Chachar [5], Faisal Hayat [1], Bilquees Bozdar [3], Filza Ansari [3], Rashid Ali [3], Lin Gong [6] and Panfeng Tu [1,*]

1 College of Horticulture and Landscape Architecture, Zhongkai University of Agriculture and Engineering, Guangzhou 510550, China; nachachar@sau.edu.pk (N.A.); wangcquanquan@163.com (C.W.); yongquanli@zhku.edu.cn (Y.L.); 13751774213@139.com (J.L.); schachar@sau.edu.pk (S.C.); maken_faisal@yahoo.com (F.H.)

2 Institute of Biomass Engineering, South China Agricultural University, Guangzhou 510642, China; nannandeng@163.com

3 Faculty of Crop Production, Sindh Agriculture University, Tandojam 70060, Pakistan; zhnshah@sau.edu.pk (Z.H.S.); 2k19-pd-95@student.sau.edu.pk (B.B.); 2k19-pd-122@student.sau.edu.pk (F.A.); 2k19-pd-286@student.sau.edu.pk (R.A.)

4 College of Natural Resources and Environment, South China Agricultural University, Guangzhou 510642, China; lshdeng@scau.edu.cn

5 College of Agriculture and Biology, Zhongkai University of Agriculture and Engineering, Guangzhou 510550, China; zs.chachar@gmail.com

6 Dongguan Yixiang Liquid Fertilizer Co., Ltd., Dongguan 523135, China; gonglin111@126.com

* Correspondence: tupanfeng@163.com

† These authors contributed equally to this work.

**Abstract:** The role of modified biochar in enhancing phosphorus (P) availability is gaining attention as an environmentally friendly approach to address soil P deficiency, a global agricultural challenge. Traditional phosphatic fertilizers, while essential for crop yield, are costly and environmentally detrimental owing to P fixation and leaching. Modified biochar presents a promising alternative with improved properties such as increased porosity, surface area, and cation exchange capacity. This review delves into the variability of biochar properties based on source and production methods and how these can be optimized for effective P adsorption. By adjusting properties such as pH levels and functional groups to align with the phosphate's zero point of charge, we enhance biochar's ability to adsorb and retain P, thereby increasing its bioavailability to plants. The integration of nanotechnology and advanced characterization techniques aids in understanding the structural nuances of biochar and its interactions with phosphorus. This approach offers multiple benefits: it enables farmers to use phosphorus more efficiently, reducing the need for traditional fertilizers and thereby minimizing environmental impacts, such as greenhouse gas emissions and P leaching. This review also identifies existing research gaps and future opportunities for further biochar modifications. These findings emphasize the significant potential of modified biochar in sustainable agriculture.

**Keywords:** nutrient management; agro-sustainability; engineered biochar; feedstock variability; nanotechnology; characterization; artificial intelligence

## 1. Introduction

### 1.1. Background

Phosphorus (P) is universally recognized as an essential macronutrient vital for plant growth and development. Its role in various plant physiological processes, such as energy transfer, photosynthesis, and nucleic acid synthesis, is well documented [1]. It also plays a pivotal role in plant food production, as it forms a fundamental component of plant DNA, RNA, and ATP and participates in critical biochemical processes vital for plant

development and reproduction [2,3]. The significance of P in agriculture can be traced back to ancient farming practices. Animal bones, rich in P, were incorporated into the soil to bolster crop yields, thereby highlighting the vital role of P in soil fertility and plant growth. Over time, methods for enhancing soil with P have diversified. Before the late 19th century, P sources such as urine, animal manure, human excreta, bone ash, and guano (seabird droppings) were extensively utilized. Today, wastewater treatment plants and animal farms are major contributors to the production of nutrient-rich materials, including sewage sludge, effluent, manure, and animal slaughter by-products like meat and bone meal, all rich in P [4].

The global population is projected to reach 9.7 billion by 2050, placing significant pressure on agriculture to enhance productivity while ensuring food security within the constraints of limited arable land. Efficient fertilizer utilization is paramount in this context. However, the economic efficiency of mineral fertilizers has declined owing to rising prices, sometimes surpassing the cost of food production. Agriculture also faces challenges posed by global climate change driven by greenhouse gas (GHG) emissions. Finite geological resources for manufacturing fertilizers and market fluctuations in fertilizer minerals intensify competition and jeopardize food security [5]. Notably, the finite geological reserves of phosphate rock, the primary source of P fertilizer, are not considered in agricultural practices or global trade, presenting a short-sighted approach. However, phosphate rock mining underpins modern agriculture, supporting the boom in human population prosperity [6].

In modern agriculture, the reliance on phosphate fertilizers sourced from depleting non-renewable phosphate rock reserves is intensifying alongside the rapidly increasing global food demand (Figure 1). This highlights the urgent need for sustainable phosphorus (P) management strategies [6]. The direct application of P to soils often results in its fixation, primarily through chemical interactions with soil minerals like iron and aluminum oxides in acidic conditions or with calcium in alkaline soils, forming insoluble compounds that limit its availability to plants [2,7]. Such fixation reduces the efficiency of fertilizers by immobilizing a significant portion of P, making it inaccessible to plants [8] and contributing to environmental degradation. Persistent use of P-enriched fertilizers and manure leads to P accumulation in soils, which, through erosion and leaching, can affect aquatic ecosystems by promoting eutrophication [9]. Moreover, the inefficiency in P use raises concerns about the future scarcity of this finite resource [10]. Given these challenges, developing innovative strategies to enhance P availability in soils is imperative, ensuring both agricultural productivity and environmental sustainability.

*1.2. P Dynamics in Soils*

As illustrated in Figure 2, P dynamics in soils underscores its integral role in plant nutrition and broader agricultural sustainability. The P cycle is a complex interplay of biogeochemical processes that regulate the movement, transformation, and availability of P in terrestrial ecosystems [7]. The cycle begins by weathering primary minerals, such as apatite, releasing P into the soil system. Once present in soil, P can undergo various transformations. It can be absorbed by plants primarily as orthophosphate ions, either as $H_2PO_4^-$ in acidic soils or as $HPO_4^{2-}$ in alkaline conditions [8]. This soil solution P serves as the direct source for plant uptake, as its reverted or fixed forms are not as accessible. However, P is not only statically held in the soil. It interacts dynamically with both organic and inorganic matter. Soil microorganisms play a role in the mineralization of organic P, converting it into inorganic forms that plants can utilize [11].

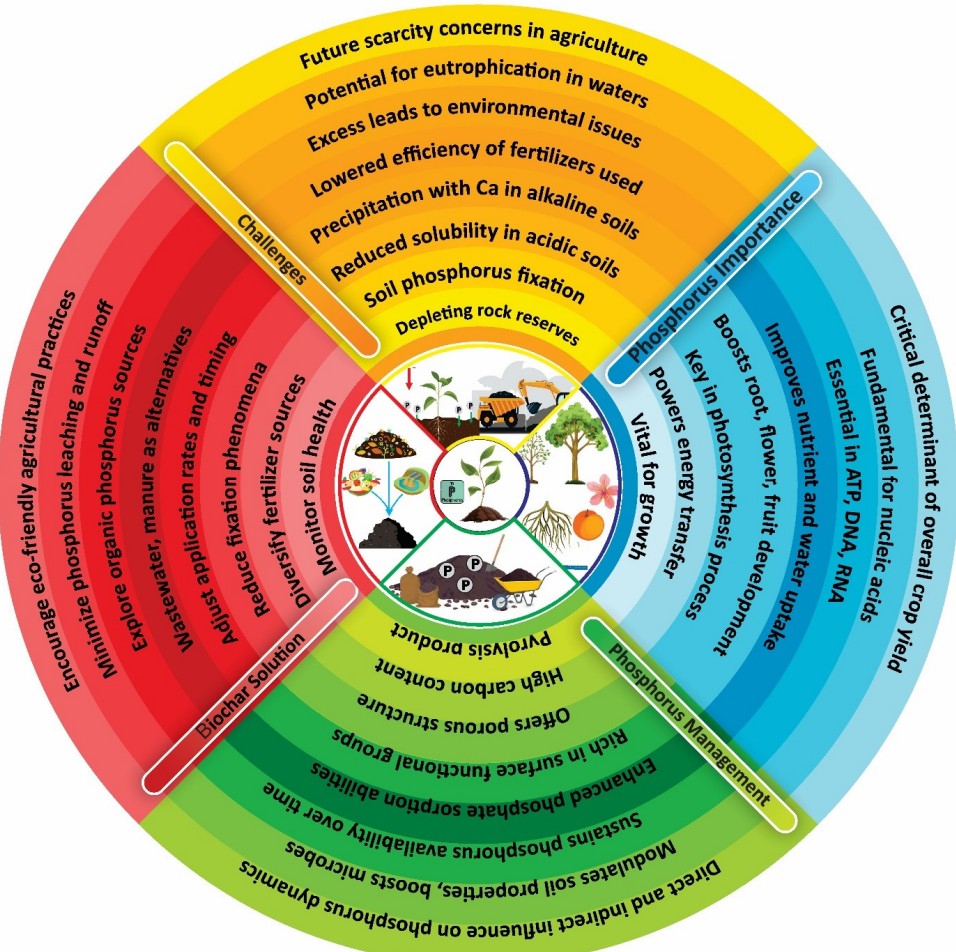

**Figure 1.** P in agriculture: its role, challenges in soil fixation, management strategies, and the potential of biochar.

Conversely, immobilization causes plants and microbes to take up inorganic P, converting it into organic form. Additionally, various reactions such as adsorption, desorption, dissolution, and precipitation determine the concentration of P ions in the soil solution. A significant consideration in the P-cycle is the potential for loss. Soil erosion and runoff can carry away both particulate and soluble forms of P, which can lead to aquatic eutrophication if they enter water bodies [12]. Leaching is another pathway through which soluble P might move to deeper soil layers or groundwater, rendering it unavailable to plants [13]. To counterbalance these losses and maintain soil P levels, external inputs such as compost, manure, biosolids, phosphatic fertilizers, and biochar are often incorporated [10,14]. These inputs undergo transformations, further contributing to the dynamic nature of the P cycle. Hence, adequate P availability in the soil solution must be ensured to achieve economically optimal crop yields.

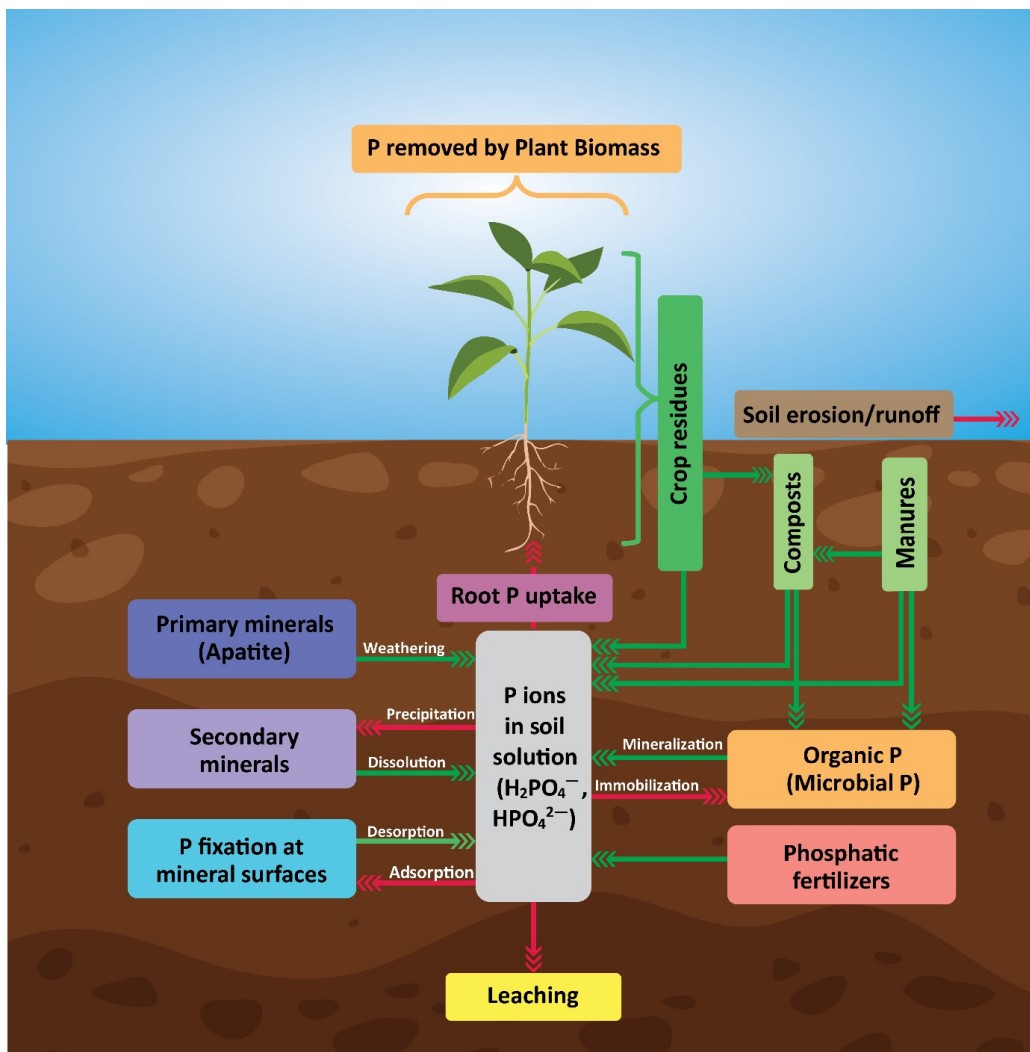

**Figure 2.** P dynamics in soils: a comprehensive depiction of sources, transformations, and losses that govern the availability and movement of P in terrestrial ecosystems.

*1.3. Overview of Biochar and Its Role in P Management*

Biochar is derived from the pyrolysis of organic materials in oxygen-limited environments. During the last few years, research on biochar has gained significant momentum in both environmental and agricultural research (Figure 1). This carbon-rich product is distinguished by its high carbon content, abundant surface functional groups, and porous structure, making it a versatile candidate for various applications [15,16]. Moreover, the potential of biochar to revolutionize P management in agriculture has come to the forefront. This versatile soil amendment possesses intrinsic properties that make it a unique solution for addressing the challenges of soil P fixation. Its highly porous structure and significant surface area, coupled with the presence of functional groups, enable biochar to effectively adsorb P [10,17,18]. Although unmodified biochar generally has low phosphate sorption capacity, mineral-rich biochar is an exception. Engineered biochar, modified with various elements, features enhanced surface characteristics such as charge, surface area, pore volume, and functionality. These modifications significantly boost its phosphate sorption capacities, turning biochar into an effective reservoir for adsorbed phosphorus, thereby ensuring prolonged availability to plants [13,17]. Additionally, certain biochars can be tailored to achieve a slow-release pattern of P, providing a sustainable source of P for plant growth [19]. The continued emphasis on research in this area underscores the potential of biochar as a sustainable solution to contemporary environmental challenges.

Furthermore, the influence of biochar on P dynamics extends beyond that of direct adsorption. It indirectly affects soil ecosystems by improving soil structure, water retention, and microbial communities (Figure 1). These enhancements foster plant nutrient uptake, increase soil nutrient availability, and promote microbial activity that assists P solubilization [20,21]. However, the efficacy of biochar as a P management tool is influenced by factors such as feedstock, pyrolysis conditions, and post-production modifications, which can significantly alter its characteristics and, consequently, its P adsorption capacity. Biochar has emerged as a pivotal solution to counter P fixation challenges in agriculture. Owing to its versatile attributes, including P adsorption, soil condition enhancement, and facilitation of microbial interactions, biochar is a potent resource for optimizing P use and reducing environmental implications. The purpose of this article is to comprehensively review recent advancements in biochar modifications aimed at enhancing phosphorus utilization in agriculture, identify the existing gaps in the research, and suggest directions for future studies.

## 2. Phosphorus Fixation Challenges and Impacts

### 2.1. Significance of Phosphorus in Crop Nutrition

P is an essential macronutrient for plant growth and development. It plays pivotal roles in various physiological processes, including photosynthesis, respiration, protein synthesis, nucleic acid formation, and energy transfer [1,2]. It is also considered integral to ATP production, which is the primary cellular energy source [22]. In addition, they contribute to the formation of phospholipids, which are essential components of cell membranes that serve as sensory interfaces and in metabolic processes [23]. The role of P in crop nutrition extends to root development, flowering, and fruiting. Adequate P levels support robust root systems, improving nutrient and water uptake [24]. Moreover, it is critical for flower and seed production during the reproductive phase and ultimately influences crop yield [25]. Despite its importance, P is often present in the soil in forms that are inaccessible to plants owing to fixation processes [7]. This limitation can lead to P deficiency in crops, resulting in stunted growth and reduced yield.

### 2.2. Detrimental Effects of P Fixation

P fixation in soil presents multifaceted challenges for sustainable agriculture, encompassing economic and environmental implications (Figure 1). Reduced crop productivity due to P deficiency resulting from fixation leads to stunted growth, decreased flowering, and yield decline, causing financial losses to farmers despite increased fertilizer use [26]. The economic repercussions are exacerbated by the necessity for increased fertilizer application rates to counteract P fixation, culminating in heightened farming expenditures devoid of guaranteed commensurate yield enhancements. These augmented agricultural costs place a substantial burden on farmers, primarily stemming from the escalated expenses incurred by intensified fertilizer utilization to combat P fixation. Consequently, these elevated costs pose a significant threat to the sustainability of farming operations, a concern that is particularly acute for smallholder farmers [27,28].

P fixation has profound environmental implications, notably leading to environmental degradation. It gives rise to the runoff of P into freshwater bodies, ultimately resulting in water pollution and eutrophication, thereby jeopardizing aquatic ecosystems. The excessive use of P fertilizers in response to fixation can exacerbate nutrient runoff into freshwater bodies, intensifying eutrophication, characterized by oxygen depletion, proliferation of harmful algal blooms, and loss of aquatic life [9,29]. Moreover, the production and transport of P fertilizers are associated with greenhouse gas emissions, intensifying the contribution of agriculture to global warming [9,30]. Unsustainable resource use is another outcome of P fixation. As the global reserves of rock phosphate, a primary P fertilizer source, are depleted, optimizing P use in agriculture is imperative. P fixation reduces the efficiency of applied P, necessitating more resource extraction to meet agricultural demands [10]. The degradation of soil health is a significant concern associated with P fixation. Continuous

P fertilizer application can disrupt soil microbial communities, reduce organic matter, and create secondary nutrient imbalances, ultimately compromising both soil health and productivity [29,31,32]. Addressing P fixation is imperative because these changes can degrade soil health over time, affecting resilience and overall capacity to sustain crops and ecosystems. It is essential not only to enhance crop productivity but also to mitigate its far-reaching economic, environmental, and sustainability challenges.

## 3. Role of Biochar in P Adsorption

### 3.1. Basic Mechanism of P Adsorption by Biochar

Biochar, which originates from the pyrolysis of organic materials in an oxygen-limited environment, has rapidly gained attention in the agricultural realm, particularly for its capacity to enhance soil fertility and mitigate P fixation. Understanding the mechanisms underpinning the ability of biochar to adsorb P provides insights into its multifaceted benefits and the means to tailor its production for specific agricultural needs. The fundamental mechanisms governing the capacity of biochar to adsorb P are based on its physicochemical attributes. The porous structure, extensive surface area, abundant surface groups, and mineral content of biochar play pivotal roles in its P adsorption capability [1]. The porous structure of biochar, characterized by its intricate network of pores, provides numerous sites for P adsorption, effectively capturing soluble P from soil solutions and preventing its loss or fixation. Additionally, the surface chemistry of biochar, enriched with diverse functional groups such as hydroxyl, carboxyl, and phenolic groups, actively interacts with P to form complexes that facilitate P adhesion to the biochar surface. Electrostatic interactions further enhance this process. Biochar surfaces often carry a negative charge, which attracts positively charged P species, such as $H_2PO_4^-$ and $HPO_4^{2-}$ [33]. This electrostatic attraction facilitates the adsorption of P onto the biochar surface.

The ability of biochar to modulate soil pH, particularly in acidic soils, contributes to P availability. By increasing the soil pH to an optimal range, biochar enhances P solubility and reduces the risk of soils [34]. Ligand exchange is another significant mechanism of P adsorption, especially for biochars that undergo post-pyrolysis modifications to introduce or enhance specific surface functionalities [34]. This process involves the substitution of other ions with P ions on the biochar surface, thereby contributing to P retention (Figure 3). Furthermore, biochar has the potential to serve as a slow-release fertilizer, gradually releasing adsorbed P over time [35]. This controlled release benefits plant growth while minimizing P runoff into water bodies [33]. In addition to its role in P adsorption, biochar can effectively address soil acidity issues and improve soil health and fertility. Various types of biochar, enriched with minerals and produced at different temperatures, have been found to increase soil pH and basic cation retention, thereby promoting plant growth and yield [34]. Diverse biochar applications have been extended to the removal and recovery of P from aquatic environments, thereby contributing to eutrophication control and sustainable P reuse in agriculture. Modified biochars have demonstrated high P adsorption capacities, effectively reducing total P concentrations in water bodies and inhibiting algal growth [12]. Overall, biochar's multifaceted properties and mechanisms make it a valuable tool for managing P in agriculture, enhancing soil fertility, and mitigating the detrimental effects of P fixation.

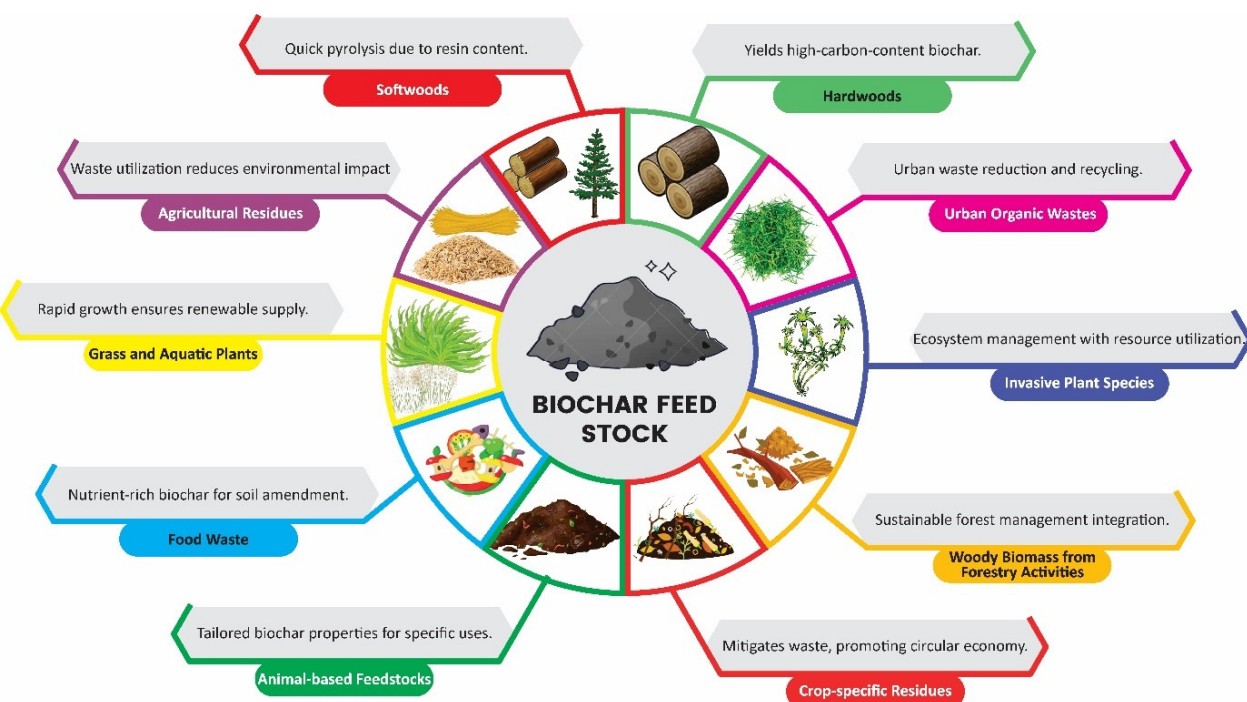

**Figure 3.** A visual representation of the diverse feedstocks and their unique contributions to environmentally beneficial and economically viable biochar sourcing.

### 3.2. Factors Influencing Adsorption and Desorption

Understanding biochar's adsorption and desorption mechanisms is essential for predicting its behavior in soil environments and optimizing its use in P management. Various factors play a role in determining the efficiency and reversibility of P binding to biochar surfaces [33,36].

**Surface area and porosity:** The micro-and mesoporosity of biochar significantly influence P adsorption. A higher surface area means more available sites for P to bind, with the pores acting as reservoirs, trapping P, and thereby modulating its availability to plants [1].

**Surface functional groups:** The chemisorption of P onto biochar is influenced by its functional groups, such as carboxylic (–COOH), hydroxyl (–OH), and phenolic (–ArOH) groups. These functional groups play pivotal roles in interactions with P ions, primarily through mechanisms such as ligand exchange and electrostatic attraction [1,37].

**The pH of the system:** The pH of the biochar and the surrounding environment plays a crucial role in adsorption. Typically, biochars are alkaline, making their surfaces negatively charged and thus facilitating the adsorption of positively charged ions such as $Ca^{2+}$, $Mg^{2+}$, $K^+$, and $NH_{4+}$ [38]. Moreover, pH affects the dominant adsorption mechanism by influencing the P species and surface characteristics [1,39].

**The presence of competing ions:** Ions such as $Cl^-$, $SO_4^{2-}$, and $HCO_3^-$ in contaminated water and soil can affect phosphate adsorption [40,41]. Moreover, ions such as Ca, Mg, and Al present in the soil can compete with P for available adsorption sites on biochar. These competing ions, especially $SO_4^{2-}$ and $HCO_3^-$, can hinder the formation of calcium and magnesium phosphate, thereby affecting the adsorption efficiency [40].

**Biochar mineral content:** The mineral content of biochar, either inherent or introduced during pyrolysis, can aid in P adsorption. Minerals such as Ca, La, Fe, and Al can bond strongly with P, enhancing its retention capacity [42,43]. For example, biochar modified with Ca from oyster shells or La showed enhanced P adsorption capabilities over a broad pH range, demonstrating the significant influence of mineral content on the adsorption mechanism [43]. Moreover, research by Yang et al. [44] and Cui et al. [42] has emphasized that composite biochars impregnated with $FeCl_3$ or $MgCl_2$ can efficiently recover P from

wastewater. Their findings highlight the importance of mineral matching in biochar to optimize P recovery and reduce secondary pollution.

**Temperature and contact time:** Both the temperature of the system and the duration of contact of the biochar with P-rich solutions influence the adsorption kinetics. Elevated temperatures can accelerate adsorption, whereas extended contact times may saturate the adsorption sites [45,46].

Biochar properties are also influenced by factors such as the pyrolysis temperature and feedstock, which impact P adsorption. Utilizing biochar for P management offers ecological and agricultural benefits; however, its adsorption capabilities vary. Understanding these factors is crucial for tailoring applications to address the P challenges in agriculture and the environment.

## 4. Advancements in Biochar Preparation

### 4.1. Feedstock Variability in Biochar Production and Its Implications

The ability of biochar to adsorb P varies significantly depending on several key factors, including feedstock choice, pyrolysis conditions, and post-pyrolysis modifications. Biochar, with its distinct properties determined by feedstock and production parameters, is a multifunctional tool with the potential to reshape agricultural and environmental landscapes. Its efficacy in P adsorption and the attributes it brings into applications are deeply rooted in the type of feedstock utilized [1]. Historically rooted in a diverse array of biomass sources, ranging from animal residues to plant materials, biochar production has always been intertwined with the inherent characteristics of the chosen feedstock (Figure 3). Elements such as lignin and cellulose from these feedstocks play a pivotal role in determining the final attributes of biochar, such as surface properties and porosity [47]. For instance, hardwood-derived biochars, renowned for their high carbon content, offer enhanced soil stability and present an increased potential for long-term P retention [48].

On the other hand, biochars sourced from agricultural residues are often enriched with nutrients due to the nutrient-rich nature of the initial biomass, as indicated by Freitas et al. [14]. Notably, due to their inherent inorganic mineral content, certain feedstocks, such as rice husks and bones, pave the way for biochars with elevated P adsorption sites, optimizing their capacity to retain P [49]. In the modern context, the versatility of biochar is further demonstrated by studies such as that of Roberts et al. [50], which brought seaweed-derived biochar into the spotlight. These biochars, characterized by their low carbon yet rich essential trace element content, offer a pH spectrum from neutral to alkaline, showcasing their adaptability across varied soil types. Given the complex interplay between feedstock types and their resultant biochars, a comprehensive grasp of feedstock variability is crucial. Such an understanding aids in tailoring biochar applications, optimizing benefits, and truly harnessing its potential in both the agricultural and environmental domains.

### 4.2. Evolution of Pyrolysis Techniques and Their Influence on Biochar Production

Biochar, a carbon-rich product, is traditionally produced using various pyrolysis techniques. The transformation of organic materials through pyrolysis has evolved significantly, with the employed techniques profoundly affecting the properties of the resultant biochar (Table 1). Historically, biochar production has primarily relied on traditional kilns, such as simple earth mounds, pit kilns, and brick kilns, mainly used in rural areas [51]. While these kilns proved cost-effective for small-scale production, they faced challenges in terms of carbonization rate, quality, and yield and also had drawbacks related to pollution, labor intensity, and land costs. Considering the limitations of traditional kilns, innovations have led to the development of enhanced versions. These improved kilns retained the essential features of their predecessors but incorporated modifications to augment biochar yield and reduce environmental impacts, effectively bridging the gap between tradition and efficiency [51,52]. The rising scale and demand for biochar have paved the way for the advent of industrial production technologies. For large-scale operations, these systems emphasize product consistency and higher throughput. The challenges and global em-

phasis on sustainability have catalyzed the emergence of the newest pyrolysis systems, prioritizing high yields with reduced emissions. These state-of-the-art systems integrate the latest research and innovative methods, such as microwave pyrolysis, often leveraging specialized reactors for precision control. Their overarching goal is to harmoniously produce high-quality biochar while minimizing the environmental footprint and setting new standards in biochar production [53].

### 4.3. Advances in Pyrolysis Techniques

Pyrolysis, a notable thermochemical decomposition process, has been the focal point of research owing to its capability to convert organic materials into valuable products, primarily biochar [52]. The attributes of the resultant biochar, especially its P adsorption and desorption properties, are heavily influenced by various factors, such as the pyrolysis method employed, temperature, and residence time [54,55]. This transformation process plays an essential role in sustainable agricultural practices, particularly in efficiently managing P [56]. Among the various pyrolysis methodologies, slow pyrolysis has historically been significant. The favored method was characterized by extended residence times and temperatures ranging from 300 °C to 700 °C [57]. This method has traditionally been associated with producing biochar using basic systems such as earth mounds, pit kilns, and brick kilns, particularly in more rural settings [57]. Conversely, fast pyrolysis, characterized by elevated temperatures and rapid heating rates, took center stage, primarily producing bio-oil with biochar and syngas as secondary outputs [55].

Recent innovations have paved the way for developing more advanced pyrolysis techniques. Microwave pyrolysis is one such method that has gained traction owing to its energy efficiency and uniform heating. This approach has emerged as a favored choice for modular systems designed for efficient solid waste management [52]. Moreover, hydrothermal carbonization has been introduced to address the challenges of processing wet biomass. This technique yields hydrochars that possess a pronounced degree of carbonization compared to their counterparts derived from torrefaction, making them uniquely suitable for specific agricultural contexts [54]. Furthermore, there have been advancements beyond traditional pyrolysis. For instance, intermediate pyrolysis reactors, also referred to as converters, are being explored for the large-scale balanced production of char and bio-oil from forests and agricultural waste without the need for exhaustive preprocessing [52]. Additionally, torrefaction, another thermal treatment, augments the qualities of biomass and biochar, enhancing their fixed carbon content and energy density rendering them valuable for energy pursuits [54]. To further advance the potential of biochar, clay–biochar composites have been developed by integrating clay minerals during pyrolysis, resulting in products with superior cation exchange capacities and P retention, potentially elevating soil quality and nutrient management [58].

Elevated pyrolysis temperatures generally result in biochar with a greater surface area, porosity, and carbon content, all of which augment its P adsorption capacity [36]. However, excessively high temperatures can lead to the volatilization of essential nutrients, potentially limiting the nutrient supplementation capacity of biochar. However, they also result in reduced CEC and volatile matter due to the extensive decomposition of organic matter [47]. Such biochars, with larger surface areas, heightened porosities, and produced at higher temperatures, have properties that can notably improve P adsorption [36]. Despite high temperatures enhancing certain properties, they can also diminish the functional groups essential for P binding [45]. Thus, the quest to determine an optimal temperature to produce biochar tailored for specific applications remains at the forefront of many studies.

The residence time, categorized into slow and fast pyrolysis, is another decisive factor in determining biochar quality and yield. Characterized by temperatures ranging from 300 to 700 °C and longer residence times, slow pyrolysis predominantly produces biochar. These extended residence times can foster secondary reactions, refine the biochar structure, and influence its P retention potential [59]. This method is favorable when producing higher biochar outputs that excel in P adsorption, as demonstrated by Tenic et al. [55]. In contrast,

fast pyrolysis, with its short residence times and elevated temperatures, is geared towards maximizing bio-oil production, relegating biochar to a secondary product [53]. Biochars derived from slow pyrolysis typically exhibit a high fixed carbon content, suggesting superior stability and microbial decomposition resistance. Such biochars, especially those sourced from lignin-rich feedstocks, have been highlighted for their enhanced P adsorption capacities, implying their suitability for long-term environmental applications [53,60]. Drawing from research findings, it is evident that pyrolysis temperature and residence time, particularly in slow pyrolysis, play a central role in determining the P adsorption and desorption attributes of biochar. The choice of these production parameters significantly shapes the physicochemical properties of biochar. Additionally, sourcing biochar from P-rich feedstocks accentuates its critical role in the ecological P cycle. Mastering these production nuances ensures that biochar remains a pivotal tool for sustainable P management.

**Table 1.** Comparison of traditional and modified biochar characteristics.

| Property/ Characteristic | Traditional Biochar | Modified Biochar | Impact on P Adsorption | P Adsorption by Modified Biochar | Experimental Conditions | References |
|---|---|---|---|---|---|---|
| **Porosity** | Low to moderate | Enhanced, due to specific modification techniques | Higher porosity can increase the surface area available for P adsorption | 620 mg g$^{-1}$ | Phosphate solutions | [61,62] |
| **Surface Area (m²/g)** | Typically <300 | Can exceed 1000, depending on the modification technique | A larger surface area provides more adsorption sites, increasing P retention | 10.4 mg g$^{-1}$ | P-containing wastewater | [63,64] |
| **pH** | Generally alkaline, but variable (6–9) | Can be fine-tuned to desired values using specific precursors or post-treatment methods | pH close to phosphate's zero point of charge (pH_zpc) can enhance P adsorption | 95.2 mg g$^{-1}$ | River sediment-water | [12,61] |
| **Cation Exchange Capacity (CEC)** | Moderate | Enhanced due to the addition of functional groups or mineral phases | Higher CEC can lead to better P retention by promoting ion exchange | 28–29 mg g$^{-1}$ | Phosphate solutions | [65] |
| **Presence of Functional Groups** | Limited presence of hydroxyl, carboxyl, and phenolic groups | Enriched with specific functional groups post-modification | Functional groups play a crucial role in P binding, especially hydroxyl groups | 24.7 mg g$^{-1}$ | Phosphate solutions | [35] |
| **Stability in Soil** | Moderate | Enhanced, especially if cross-linked or treated with minerals | Stable biochars persist longer in soil, providing sustained P management | Reduced P runoff from soil; greater microaggregate stability. | Temperate Agricultural Soil | [66] |
| **Hydrophobicity** | Often high due to carbon-rich nature | Can be adjusted using post-treatments | Lower hydrophobicity may promote aqueous interactions and P adsorption | 56.12 mg g$^{-1}$ | Phosphate solution | [67,68] |
| **Metal Content** | It depends on the biomass source | It can be enriched if treated with metal solutions | Metals can act as bridges for P, enhancing its adsorption onto biochar through ligand exchange and electrostatic attraction | 19.66 mg g$^{-1}$ | Phosphate solution | [64,69] |

Comparison of the properties and characteristics of traditional and modified biochars and their implications for P adsorption. The table provides an overview of how specific modifications in biochar can enhance its efficiency in P management.

## 5. Modification and Characterization

### 5.1. Biochar Modification Techniques

Post-pyrolysis modification of biochar can significantly boost its ability to adsorb P. Utilizing activation agents such as steam or carbon dioxide augments the biochar's surface area and microporosity. The integration of nanoparticles, metalloids, and alterations in functional groups further optimized P retention capabilities (Figure 4). Modern enhancements in biochar modification techniques are paving the way for improved performance of this carbon-rich material, particularly in agricultural and environmental contexts. By tailoring the properties of biochar to fit specific needs, its efficacy in nutrient and water retention and contaminant immobilization is greatly amplified.

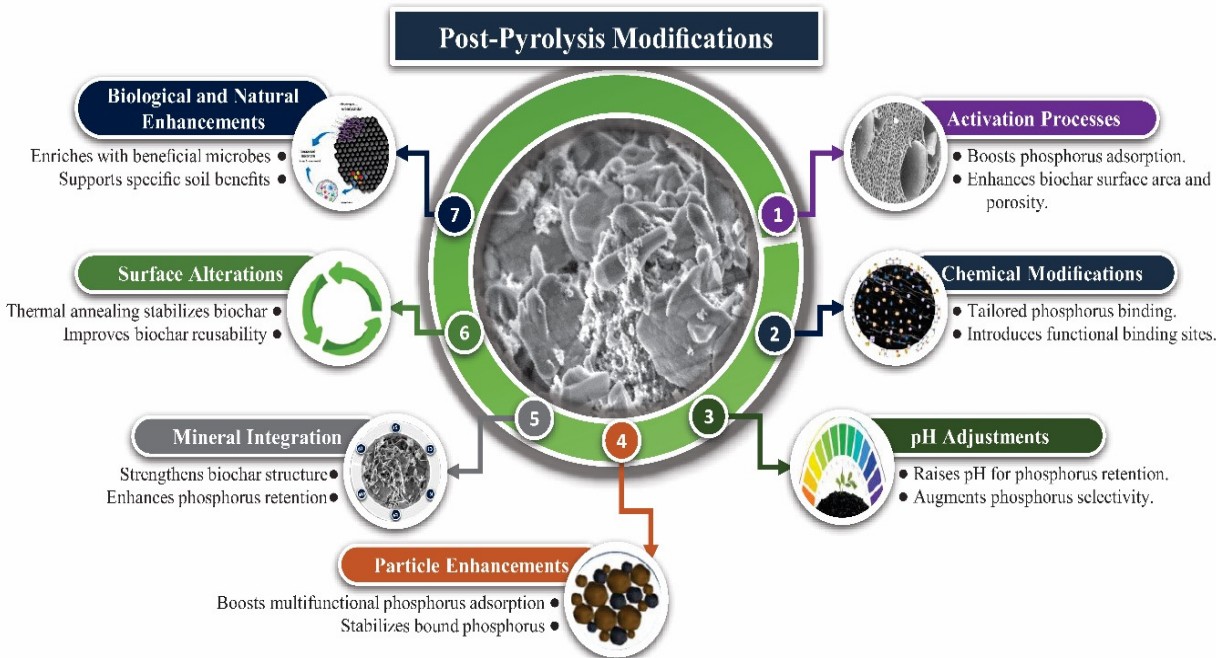

**Figure 4.** A comprehensive overview of post-pyrolysis modifications for biochar: enhancing structure, functionality, and P retention capabilities.

### 5.2. Functional Groups in Biochar

Biochar has garnered increasing attention for environmental applications, especially because of its rich functional groups that play a vital role in adsorption and pollutant remediation. Among these alkaline functional groups, predominantly hydroxyl (–OH) and carboxylate ($–COO^-$) groups are essential for their pronounced influence on P adsorption and retention of other ions [70,71]. The abundance and density of these alkaline functional groups on the biochar surfaces varied significantly depending on the feedstock and pyrolysis conditions. For example, a higher pyrolysis temperature often leads to the formation of more aromatic structures, thereby reducing the density of these functional groups. However, moderate pyrolysis temperatures may enhance the concentration of alkaline functional groups and optimize P adsorption [72]. The influence of different temperatures and carrier gases on biochar yields and properties revealed that higher temperatures result in fewer functional groups and a more significant surface area [72]. Modification of post-production processes, such as chemical activation or functionalization, can further introduce or enhance alkaline functional groups (Figure 4). Chemical treatments using strong alkalis, such as KOH or NaOH, can considerably increase the number of hydroxyl and carboxylate groups [71,73]. Owing to its tailored functionalities, biochar has emerged as a promising material for remedying soils laden with heavy metals, reducing phosphorous fixation, and improving soil properties [74]. The application of novel biochar materials, such as ball-milled P-loaded biochar (BPBCs) prepared by combined ball milling and P loading,

has been observed to enhance soil properties and increase soil nutrient concentrations. BPBCs also help reduce soil alkalization and promote plant growth in coastal saline-alkali soils [70]. Moreover, these electrostatic interactions, driven by the negative charges of the alkaline functional groups, ensure P retention on the biochar surface, thereby providing a solution for reducing water pollution [75].

Chemical modifications, such as the introduction of Mg, Ca, K, Fe, Zn, or Al, can also enhance the affinity of biochar for P (Figure 4). Moreover, modification of biochar through processes such as Mg impregnation can further enhance its adsorption capacity for P. For instance, hardwood biochar modified with Mg exhibited a 34% increase in its adsorption capacity, making it a promising candidate for phosphate recovery and subsequent slow-release fertilizer applications [76]. One notable breakthrough is chemical activation, in which biochar is treated with activating agents such as potassium hydroxide (KOH), phosphoric acid ($H_3PO_4$), or zinc chloride ($ZnCl_2$). This treatment significantly increases biochar's surface area and porosity, thus improving its adsorption capabilities, particularly for heavy metals, organic pollutants, and essential nutrients such as P [77]. Furthermore, iron oxides are recognized for their strong affinity to phosphate, making them ideal candidates for biochar modification. A study conducted by Wu et al. [41] explored rice straw-derived biochar modified with ferrous chloride (Fe(II)) and ferric chloride (Fe(III)). Notably, the Fe(II) biochar displayed superior phosphate adsorption capabilities, attributed to its amorphous state of FeOOH, which has high isoelectric points. The presence of Fe enhances phosphate adsorption through mechanisms such as electrostatic attraction and ligand exchange. Moreover, these field experiments highlighted that chemically modified biochars boosted available P and remarkably decreased leaching in saline-alkaline soils.

Calcium, which is abundant in various waste materials, is another pivotal element in P chemistry. Utilizing marble waste and calcium-rich sepiolite, Deng et al. [78] synthesized Ca/Mg-biochar composites that demonstrated exceptional phosphate adsorption. The mechanism is driven by reactions where Ca or Mg ions in the biochar react with phosphate to form precipitates such as $Ca_5(PO_4)3OH$ and $Mg_3(PO_4)_2$, mitigating phosphate mobility. In a recent study by Tu et al. [79], the research focused on the effectiveness of biochar modified with MgO for P recovery. This study involved the co-pyrolysis of MgO with various raw materials, resulting in different modified biochars: MgO–rice straw, MgO–corn straw, MgO–Camellia oleifera shells, and MgO–garden waste. The results demonstrated a significant improvement in the P adsorption capacities of these modified biochars, with MgO–rice straw displaying the highest capacity. The mechanisms responsible for P adsorption were identified as physical adsorption, precipitation, and surface inner-sphere complexation, with electrostatic attraction playing a limited role.

Additionally, the study found that P adsorbed on these biochars could be released under various pH conditions. MgO–rice straw exhibited modest desorption efficiency, making it a potential candidate for slow-release fertilizers [35]. Extending the exploration to lanthanum, Feng et al. [43] investigated a calcium-modified biochar incorporating sheep manure and oyster shells, showing a limitation in its low-pH adaptability. To counteract this, lanthanum was integrated into biochar, resulting in consistent phosphate adsorption across a wide pH range. The distribution of calcium and lanthanum in the biochar matrix, predominantly on the surface and internal pore structure, respectively, is pivotal for its performance. Another promising avenue for biochar modification is co-pyrolysis, which involves blending biomass with inorganic or organic additives during pyrolysis. For example, combining livestock manure or algae with biomass during pyrolysis allows for the customization of the nutrient profile of the resulting biochar, making it a potent and tailored fertilizer for agricultural use [80]. Therefore, algal-derived biochar is a valuable resource. Rich in nutrients and boasting strong ion-exchange capacity, algal biochar has applications in agriculture, acting as a cost-effective and efficient fertilizer, and in wastewater treatment owing to its porous structure and ion-exchange capabilities [81]. Recognizing the essential role of alkaline functional groups in biochar enables the tailored creation of biochar primed for P adsorption. Modified biochars, enriched with metals such

as Mg, Ca, K, Fe, Zn, Al, and La, present innovative solutions for soil contamination and P sequestration, promoting sustainable agriculture while repurposing waste materials. This amplifies biochar's efficiency and addresses the pressing issue of P mobility in soils.

*5.3. Nanotechnology and Its Role in Enhancing P Adsorption*

The potential of nanotechnology in the field of environmental sustainability has been captured by its application to biochar. Nanomaterials (NMs), which are central to nanotechnology, exhibit unique attributes, such as large surface areas, superior cation exchangeability, and heightened ion absorption capabilities [82]. Nanocomposite biochars have gained attention by leveraging nanotechnology to introduce nanoparticles into the biochar matrix, such as metal oxides or bio-based nano-compounds. These nanocomposites exhibit targeted functionalities, rendering them effective for precise nutrient release and pollutant remediation [83]. The nuanced differences between NMs and their bulk counterparts can revolutionize how contaminants are addressed, particularly in P adsorption. Yuan et al. [84] demonstrated that owing to their minuscule size and adaptable surface chemistry, nanoscale materials can achieve a more intimate level of interaction with P. When harnessed in biochar, this translates to augmented surface area testing with active sites optimized for P adsorption, leading to enhanced adsorption rates. Research in this domain is dynamic and diverse.

Researchers have made significant efforts to develop nanoscale biochar solutions. For instance, Sun et al. [85] reported a project that resulted in a nano-biochar composite formulated using nanoSiO$_2$ doping. This composite shone in purifying P-rich waters and proved its poor adsorption capacity, recyclability, and environmental compatibility when pitted against more traditional straw biochar materials. Wu et al. [41] ventured into nanoparticle–biochar integrations with a spotlight on iron oxide. Their rigorous experiments, which utilized rice straw-derived biochar infused with ferrous chloride (Fe(II)) and ferric chloride (Fe(III)), yielded compelling results. Fe(II) biochar has emerged as a frontrunner in phosphate adsorption and has displayed robust resilience against environmental challenges, such as pH shifts and competing anions. This was not just a laboratory victory; the Fe(II) biochar exhibited an 86.4% reduction in leaching, demonstrating its real-world application in mitigating P losses, especially in saline-alkaline soils.

Delving deeper into the realm of P adsorption, Cui et al. [86] demonstrated the advantages of FLO@CSL as a novel adsorbent. This brainchild, which is a FeLaO$_3$-modified sulfomethylated lignin (SL) biochar, hinges on the synergistic affinity of lanthanum (La) and iron (Fe) (hydro) oxides for phosphate. Its standout features include remarkable adsorption capacity and a streamlined magnetic separation process, making it a potential game-changer in wastewater treatment. However, the innovation odyssey continues. Yin et al. [87] investigated the challenge of water eutrophication, shedding light on the crucial role that modified biochars can play. Their extensive experimentation led them to pinpoint Mg–Al-modified biochars as a formidable solution, especially when confronted with complex mixtures, such as the coexistence of $NH_4^+$, $NO_3^-$, and $PO_4^{3-}$. Peng et al. [88] introduced new frontiers by focusing on the agricultural sector. Their work on metal oxide-modified biochars, especially those bolstered by FeAl and MgAl, revealed multifaceted benefits. These ranged from heightened soil P availability and promotion of inorganic P-solubilizing bacteria to a marked reduction in P leaching. Advancements in biochar research were also enriched by Zhang et al. [89] with their nano zero-valent zinc (nZVZ), which aimed to enhance the active sites, thereby improving P adsorption capacity.

Furthermore, Ce$^{3+}$-enriched ultrafine ceria nanoparticle-loaded biochar exhibited a rapid and efficient phosphate adsorption capacity, which is particularly beneficial owing to the unique characteristics of ceria nanoparticles [90]. In salt-affected soils, nano-biochar amendments have shown the potential to enhance P adsorption due to oxygenated functional groups [91]. Nano zero-valent iron-modified biochar has demonstrated heightened phosphate adsorption capacities, proving especially effective in eutrophication control and potential agricultural applications [12]. Lastly, nano-MgO biochar composites have

been identified as potent adsorbents, with their efficacy amplified when the biochar is co-pyrolyzed with magnesium citrate, showcasing impressive P immobilization [92]. Collectively, these studies highlight nanotechnology's transformative role in amplifying biochar's P adsorption capacity. This synergy promises environmental advancements and paves the way for enhanced agricultural productivity. As we navigate these innovations, ensuring a future where agriculture seamlessly blends productivity with environmental mindfulness is paramount.

## 6. Biochar in Sustainable Agriculture

Biochar, a carbon-rich material produced from the pyrolysis of biomass, offers a holistic and sustainable solution to various agricultural challenges in addition to traditional soil amendment methods [55]. The conversion of crop residues into biochar can sequester large amounts of $CO_2$ with substantial potential in specific regions. One of its key advantages is that, when incorporated into agricultural practices, it offers significant carbon sequestration potential at the national level, contributing to climate change mitigation [93]. Studies have demonstrated the capacity of biochar to sequester carbon from the atmosphere, with potential variations based on soil pH levels. Notably, acidic soils tend to release more $CO_2$ after biochar application than neutral or alkaline soils, emphasizing the need to consider soil pH when assessing carbon sequestration potential [94]. Additionally, the use of biochar in agriculture can mitigate various environmental issues, including marine aquatic biodiversity destruction, soil and water acidification, and eutrophication [95]. The inherent minerals present in biomass significantly influence carbon conversion during pyrolysis and, consequently, the properties of biochar. Removing these minerals before pyrolysis has increased carbon retention in biochar and enhanced its stability. This removal process produces biochar with a higher chemical and thermal oxidation decomposition resistance, making it a more effective carbon sequestration tool [96].

In terms of soil benefits, the porous structure of biochar enhances soil porosity and aggregate stability, leading to improved water dynamics. This enhances water infiltration and retention, making it particularly valuable in arid regions [97]. Additionally, the cation exchange capacity of biochar helps retain essential nutrients, such as ammonium, nitrate, and phosphate, thereby reducing nutrient runoff and its associated ecological impacts [37,87]. Biochar also provides a conducive habitat for beneficial soil microbes, thereby increasing microbial diversity and metabolic activity. This microbial enhancement can further boost soil health and agricultural productivity [37]. Furthermore, biochar is instrumental in immobilizing contaminants, including heavy metals such as $Cu^{2+}$, $Cd^{2+}$, and $Pb^{2+}$, as well as organic pollutants. This action prevents these contaminants from entering the food chain and enhances plant growth. Modified versions of biochar, such as those enhanced with chitosan, demonstrate improved removal of heavy metals from solutions and diminished metal toxicity in soils [98,99]. Economically, although the initial cost of biochar may be higher than that of conventional fertilizers, its long-term impact on soil health and reduced need for recurrent fertilizer applications make it a cost-effective choice. The sustained benefits of a single biochar application on soil health outweigh the costs of routine fertilizer applications. Additionally, the role of biochar in mitigating environmental issues such as nutrient leaching and runoff can result in long-term economic and ecological benefits [100]. As global agriculture shifts towards sustainable practices, biochar's multifaceted benefits become increasingly evident, surpassing traditional soil amendment methods [93]. Biochar is a versatile and sustainable solution that addresses numerous agricultural challenges, from carbon sequestration to soil enhancement, nutrient preservation, bolstered microbial activity, and contaminant immobilization.

### 6.1. Integration of Advanced Analytical Methods with Cutting-Edge Characterization Techniques and Their Insights

Biochar, derived from the thermal decomposition of organic materials, is a promising solution to various environmental challenges. Its adaptability depends primarily on its

physicochemical properties, which can be modified through specific pyrolysis conditions and feedstock selection. A profound understanding of both its surface attributes and inner configuration is essential for harnessing the full potential of biochar and engineering it for precise applications (Table 2). This calls for emerging advanced spectroscopy methods with state-of-the-art characterization techniques [91]. The relationship between biochar and P, especially their adsorption and desorption behaviors, is better understood because of the development of intricate characterization methodologies. These contemporary analytical instruments have enabled researchers to explore the molecular and microscopic interplay between biochar and P in detail.

**Density functional theory:** Recent studies have focused on the adsorption of phosphate ($H_2PO_4^-$) in water by metal-modified biochar. Using density functional theory, Yin et al. [101] demonstrated that metal-modified biochar exhibited a stronger molecular-level effect on phosphate adsorption than unmodified variants. In particular, Ca-modified biochar was superior to its Mg-modified counterparts. This study revealed that metal adsorption, primarily through electrostatic attraction, outperforms edge adsorption, which relies on covalent bonding.

**Imaging techniques—SEM and EDX:** The combined power of scanning electron microscopy (SEM) and energy-dispersive X-ray spectroscopy (EDX) has become indispensable. Wang et al. [90] employed these techniques to obtain high-resolution images of biochar surfaces and performed elemental mapping to locate adsorbed P and elucidate adsorption hotspots on the biochar. Furthermore, they prepared biochar-loaded $Ce^{3+}$-enriched ultrafine ceria nanoparticles (Ce-BC), which showed significant potential for phosphate removal from water.

**Surface chemistry analysis—XPS:** X-ray photoelectron spectroscopy (XPS) is a formidable technique for discerning biochar surface chemistry. Identifying the types and concentrations of functional groups involved in P binding is particularly important, as shown by Bolton et al. [102]. Their study elucidated the formation of iron phosphate and revealed that P capture is associated with various mineral phases.

**Functional group identification—FTIR:** Fourier-transform infrared spectroscopy (FTIR) has become essential for recognizing specific functional groups and chemical bonds on the biochar surface that interact with P. Several studies, including those by Liu et al. [103], Shin et al. [104], and Mahmoud et al. [91], demonstrated the capabilities of FTIR to highlight fundamental adsorption mechanisms and interactions.

**Speciation and interaction analysis—NMR:** Nuclear magnetic resonance (NMR) spectroscopy, particularly solid-state $^{31}$P NMR, offers crucial insights into the nature of P within the biochar matrix. Amin et al. [105] and Sacko et al. [106] employed NMR and other techniques such as ssNMR, XRD, and NEXAFS to elucidate how P binds to biochar and the specifics of these interactions.

The arsenal of advanced characterization techniques has exponentially augmented our understanding of the interplay between biochar and P. By unraveling the underlying mechanisms. These tools can empower researchers to refine biochar properties and set the stage for superior P management strategies in soils.

**Table 2.** Overview of advanced analytical techniques in biochar research.

| Analytical Technique | Principle/Methodology | Benefits in Biochar Research | Challenges/Limitations | References |
|---|---|---|---|---|
| **X-ray Photoelectron Spectroscopy (XPS)** | Measures the elemental composition and electronic state of elements | Reveals surface chemistry and potential functional groups | Limited to surface analysis; time-consuming | [107] |
| **Scanning Electron Microscopy (SEM)** | Provides detailed images of biochar surfaces using electron beams | Visualizes microstructure and porosity; aids in determining biochar's physical properties | Requires gold or carbon sputter coating for some samples, potentially altering surface | [108] |

**Table 2.** *Cont.*

| Analytical Technique | Principle/Methodology | Benefits in Biochar Research | Challenges/Limitations | References |
| --- | --- | --- | --- | --- |
| Fourier-Transform Infrared Spectroscopy (FTIR) | Measures vibrational frequencies to determine chemical compounds | Identifies functional groups and organic components | Limited sensitivity for very low-concentration species | [109] |
| Nuclear Magnetic Resonance (NMR) | Utilizes nuclear spins in a magnetic field | Offers insights into biochar's carbon types and distribution | Requires high concentrations of samples; relatively expensive | [110] |
| Thermogravimetric Analysis (TGA) | Monitors weight change in a material as a function of temperature or time | Assesses thermal stability and organic content of biochar | Does not provide specific information on biochar's chemical structure | [111] |
| Brunauer–Emmett–Teller (BET) Method | Measures gas adsorption on solid surfaces | Evaluates specific surface area, aiding in understanding adsorption capacity | Limited to certain gas–solid systems; does not consider pore geometry | [112] |

Overview of key analytical techniques employed in biochar research, detailing their principles, advantages, limitations, and relevance in assessing and understanding biochar structure and properties.

### 6.2. Role of Artificial Intelligence in Data Analysis and Prediction

Biochar, a carbon-rich derivative obtained by the thermal decomposition of organic materials, has gained significant attention in environmental research. This attention is primarily owing to its potential to offer solutions to some of the pressing environmental challenges. An essential facet of biochar research pertains to its intricate relationship with P, particularly concerning its adsorption and desorption behavior [113]. Understanding the nuances of this relationship requires advanced tools and methodologies. Although useful, traditional analytical methods often grapple with the intricacies of biochar-related datasets. Artificial intelligence (AI), with its sophisticated algorithms, machine learning (ML), and deep learning (DL) capabilities, has revolutionized the domain of biochar research. Notably, AI, when combined with rich historical data, can provide insights into biochar behavior, even predicting the reactions of yet-to-be-produced biochar types with P [114].

Machine learning, a subfield of AI, employs past data to train computational models. For instance, the utility of algorithms, such as neural networks, extreme gradient boosting, and random forests, in predicting biochar behavior, specifically its adsorption patterns, has been documented [113]. Furthermore, advancements in computational chemistry combined with ML have paved the way for the development of biochar as a sustainable alternative to traditional fertilizers. This is exemplified in studies aimed at creating biochar formulations capable of slow and efficient nutrient release, with an emphasis on P [115]. The integration of AI into the optimization of biochar production is particularly noteworthy. Through neural networks, it is now possible to predict the optimal feedstock and pyrolysis conditions tailored for specific outcomes, effectively transforming a traditional iterative process into a targeted data-driven approach [114]. In wastewater remediation, another realm where biochar holds promise, AI techniques such as ML and DL come into play. These techniques can predict effluent P levels even when data are scarce, facilitating compliance with regulatory standards while potentially reducing costs [116].

### 6.3. Highlighting More Nuanced AI Applications in Biochar Research

**Engineered biochar design using AI:** Liu et al. [117] utilized the power of the random forest algorithm to delve into the realm of As adsorption in Fe-modified biochar. Such applications underline the immense potential of AI in aiding the rational design of biochars, specifically tailoring them for targeted contaminant removal, such as arsenic.

**Modeling P adsorption:** A significant aspect of biochar research is its interaction with P. Tree-based AI algorithms, such as RF, DTs, and XGBoost, have made notable strides. These algorithms, particularly RF, have emerged as pivotal tools for predicting phosphate

adsorption patterns and guiding researchers in their quest to design optimal biochar-based adsorbents [118].

**Innovative hybrid models:** One of the strengths of AI is its adaptability and ability to be integrated with various computational models. This flexibility was displayed in the SVM-ANN ensemble model, which was designed to predict heavy metal sorption efficiency. By considering a plethora of variables ranging from environmental conditions and biochar physicochemical characteristics to contaminant types, AI-driven hybrid models can enhance the accuracy and scope of predictions. Such models are particularly significant for forecasting how biochar behaves under real-world conditions [114].

**Optimizing biochar production:** The potential of AI is not limited to post-production analyses. Neural networks, another facet of AI, are instrumental in predicting the optimal conditions and raw materials (feedstock) required for biochar production. Through these predictions, the traditional trial-and-error method for biochar production can be streamlined, ensuring that the desired properties of the end product can be achieved with greater efficiency [114].

**AI in wastewater treatment:** Beyond solid contaminant interactions, biochar shows promise for wastewater remediation. In this sector, ML and DL models are paramount. For instance, scientists leveraging the power of AI have predicted effluent P levels, even in incomplete datasets. Such applications of AI can revolutionize wastewater treatment processes, ensuring that the treated water complies with environmental standards while potentially minimizing treatment costs [116].

**Advanced machine learning techniques for yield prediction:** The seamless integration of AI with optimization techniques has led to the creation of predictive models, such as the ensemble learning tree (ELT-PSO). Such models have shown exemplary prediction accuracy for biochar yield, thereby minimizing the need for resource-intensive experiments [119].

The potential of AI, particularly in engineered biochar production for efficient P adsorption and desorption, augments its capability to address environmental challenges and ensure more sustainable solutions [120]. Incorporating these advanced techniques and insights ensures that biochar research is comprehensive and resource-efficient. Combining biochar technology with AI provides an enhanced understanding that is vital for effectively addressing current environmental challenges.

## 7. Practical Implications and Benefits

### 7.1. Addressing the Environmental and Health Impacts of P Leaching through Sustainable Agricultural Practices

The widespread use of P fertilizers in modern agriculture has led to significant environmental and health challenges. Notably, the leaching of excess P from agricultural terrain into adjacent water systems has myriad adverse ecological consequences [24]. Compounding this issue, intricate edaphic processes often result in the immobilization of P in the soil, thereby hindering its uptake by plants. Consequently, the efficiency of water-soluble P fertilizers remains a challenge, culminating in profound environmental and public health concerns [24]. Eutrophication of freshwater and marine ecosystems is one of the most important ecological implications of P leaching. This process, exacerbated by agricultural, urban, and industrial activities, results in an excessive influx of nutrients, primarily nitrogen and P, facilitating the uncontrolled proliferation of algal populations. As these algal blooms decay, they deplete dissolved oxygen levels, engendering hypoxic or anoxic conditions. Such environments are calamitous for aquatic life, leading to widespread fish mortality and biodiversity loss. The ramifications of eutrophication are not merely ecological; the economic impact is also substantial, with estimates indicating annual losses of $1 billion for European coastal waters and $2.4 billion for American lakes and streams [22]. Even more disconcerting are certain algal blooms dominated by cyanobacteria and blue-green algae. These blooms produce a spectrum of toxins, collectively termed cyanotoxins, which pose considerable threats to aquatic life and human health. Cyanobacterial harmful algal blooms

(CyanoHABs) severely compromise water quality. Ingestion of water or aquatic organisms, such as fish, contaminated with cyanotoxins can have deleterious health effects. Given the global surge in CyanoHAB events primarily attributed to anthropogenic eutrophication and climatic change, there is a pressing need for effective management strategies to safeguard public health and aquatic ecosystem integrity [22,121].

Biochar has emerged as a potential source of sustainable solutions. Owing to its adsorptive properties, biochar serves as an effective P sink and significantly reduces its leaching potential (Figure 5). This assertion is corroborated by empirical studies that have shown the potential of biochar to reduce P leaching by up to 60% [12,122,123]. Beyond the immediate environmental benefits, incorporating biochar into agricultural landscapes reduces the dependency on phosphate fertilizers, thus attenuating the carbon footprint associated with their manufacture and deployment [124]. With strategic measures, including the integration of biochar and a comprehensive grasp of plant physiological processes, we can achieve a harmonious balance that ensures both agricultural productivity and the protection of ecosystems and communities.

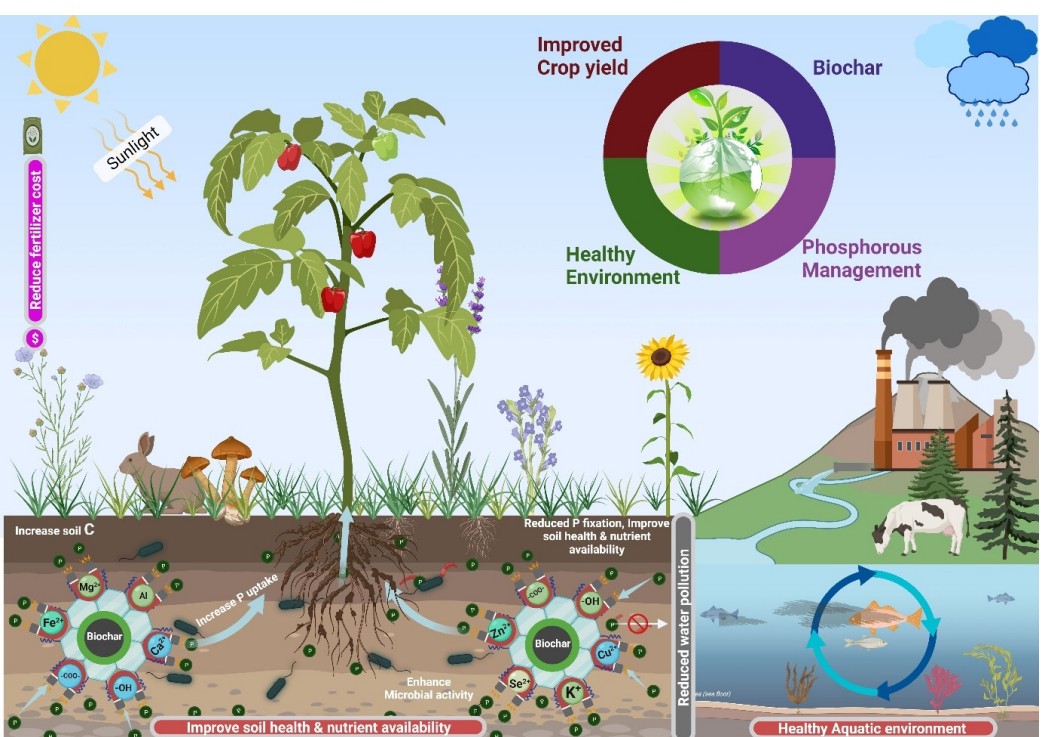

**Figure 5.** An illustrative overview of the role of biochar in P adsorption and desorption: from enhancing soil nutrient availability and microbial activity to reducing P leaching, leading to improved crop yield, a healthier environment, and mitigated aquatic eutrophication.

### 7.2. Economic Advantages in Agriculture: The Promise of Biochar

In the ever-changing landscape of agriculture and horticulture, there is an urgent need for solutions that strike a balance between economic feasibility, environmental conservation, and efficiency. Introducing biochar in this milieu can provide significant economic benefits to farmers, agronomists, and horticulturists. One primary advantage of farmers is their potential to reduce costs. The incorporation of biochar can significantly reduce the need for expensive P fertilizers. This is because of the ability of biochar to absorb and gradually release P, thereby ensuring a consistent nutrient source for crops. This reduced reliance on fertilizers could lead to substantial savings, especially for expansive agricultural ventures, as underscored by studies by Sun et al. [81] and Luo et al. [1]. Moreover, the possibility of enhanced crop yields presents a secondary yet paramount benefit. The positive link between biochar use and improved crop outcomes has been well documented, with a

notable mention being made by Li et al. [125]. Greater yields inevitably result in heightened revenue for farmers.

For agronomists and horticulturists, these implications extend beyond agricultural output. The nuanced knowledge needed to perfect biochar application, from production to utilization, could spawn specialized consulting services. Such services can guide farmers to customize biochar applications according to specific crop and soil conditions, thereby creating new employment opportunities and research directions, as highlighted by Rogers et al. [126]. Research has also highlighted the potential of combining biochar with other organic materials. For instance, Antonious et al. [127] emphasized the benefits of amalgamating animal manure and other natural substances with biochar. Such synergies can further enhance soil quality and crop yields and reduce costs, making them lucrative for farmers with limited resources. The potential of biochar as a soil revitalizer is also noteworthy, especially in areas with soil degradation. By rejuvenating soil health and its overall structure, biochar ensures sustainable cultivation for prolonged durations, circumventing the requirement for pricey soil amendments or shifting to more fertile terrain. Bista et al. [128] and Wali et al. [129] highlighted the soil-enhancing capabilities of biochar and its associated benefits for crops.

## 8. Gaps and Future Directions

### 8.1. Existing Research Gaps in Biochar Preparation and Modification

Despite the recognized potential of biochar as a sustainable soil amendment, several research gaps persist, especially in its preparation and modification techniques. A notable gap lies in the comprehensive understanding of the influence of diverse feedstocks on biochar physicochemical properties. Although investigations have been made into common feedstocks such as wood and agricultural residues, knowledge of unconventional feedstocks, such as algal biomass and sewage sludge, is limited. In particular, algal biomass is distinct from lignocellulosic biomass because of its unique surface functional groups and the presence of various cations, making it potentially effective for environmental decontamination [130].

Another significant area of research is the optimization of pyrolysis conditions based on specific feedstocks. The current literature often provides generalized pyrolysis conditions, which may not be optimal for all feedstocks. For instance, biochars from animal litter and solid waste show distinct properties compared to those derived from crop residues and wood biomass, even under the same pyrolysis conditions, mainly because of variations in lignin, cellulose, and moisture content in the biomass [47]. The post-production modification of biochar is an emerging research domain with ample room for investigation. Although some studies have investigated acid or alkali activation, the exploration of other potential modification techniques has not been exhaustive. For example, innovative methods to amplify the P adsorption capacity or longevity of biochar have yet to be thoroughly examined. One promising strategy involves the use of zero-valent iron (ZVI) biochar composites modified with $CaCl_2$ to enhance their lifespan and P removal efficacy [131]. Finally, the scalability of biochar preparation and modification techniques remains a challenge. Processes effective at the laboratory scale may not be as efficient when scaled up, hindering the economic feasibility and widespread adoption of biochar use. Although biochar holds immense promise, addressing these research gaps can pave the way for innovative applications and widespread adoption in sustainable agriculture and horticulture.

### 8.2. Opportunities for Further Innovation and Research

The utilization of biochar in agriculture is on the cusp of significant advancements, and a wide spectrum of research opportunities needs to be explored. One of the most promising research areas is the synergy between biochar application and sustainable agricultural practices. As highlighted by Elkhlifi et al. [132], the interaction of biochar with practices such as agroforestry, conservation tillage, and cover cropping can yield holistic strategies that maximize soil health and environmental benefits. Variances in biochar properties,

which depend on factors such as feedstock, pyrolysis conditions, and residence time, require further investigation to optimize their contribution to soil health, microbial activity, nutrient retention, and carbon sequestration.

Another dimension of research, as pointed out by Gillingham et al. [133], is the magnetization of biochar. This novel approach offers a dual solution for waste management: utilizing agricultural waste for magnetic biochar synthesis and facilitating nitrogen pollution management. The potential of magnetic biochar to mitigate nitrogen pollution in soils and recycle it as a fertilizer is particularly compelling, especially considering the widespread environmental and economic concerns surrounding nitrogen runoff. Tan et al. [134] highlighted the intricate relationships between biochar, soil, and microbial communities. The physicochemical properties of biochar play a critical role in shaping microbial interactions, influencing soil fertility and plant growth. Such interactions could be invaluable for the remediation of pollutants, soil enhancement, and bolstering plant resistance against pathogens. With rapid advancements in technology, there is an exciting intersection between biochar research and digital tools. Dehkordi et al. [135] proposed that integrating high-resolution imaging from UAVs with satellite data could offer deeper insights into the impact of biochar on evapotranspiration across agricultural landscapes. Furthermore, as discussed by Shaikh et al. [136], merging AI, IoT devices, and robotics with traditional agricultural practices could revolutionize biochar applications, offering precise data analysis and optimization opportunities.

Yang et al. [137] introduced the concept of the circular economy in biochar research. It is vital to explore biochar's role in broader systems, such as waste management and energy production, particularly its potential for carbon capture and storage. This could redefine the significance of biochar in both economic and environmental contexts. Finally, given the escalating concerns regarding climate change, Kumar et al. [138] suggested the potential of biochar to bolster soil resilience to extreme weather events. Understanding how biochar can mitigate the impacts of droughts, floods, or heatwaves by enhancing soil properties, such as moisture retention and aeration, could offer strategies to insulate food systems from climatic adversities. For these research opportunities to bear fruit, fostering multidisciplinary collaborations, amplifying funding avenues, and advocating for open-source data sharing is paramount. However, although this quest is challenging, the vision of a resilient and sustainable agricultural future underscores its importance.

*8.3. Potential Challenges and Solutions in Biochar Research and Application*

**Heterogeneity in biochar properties:** Biochars exhibit a vast range of physicochemical properties depending on their feedstock and pyrolysis conditions, making it difficult to consistently predict their effects on soils. The intrinsic molecular composition of biochar-derived dissolved black carbon and its interactions with metal ions, which can affect its environmental impact, are not fully understood [139]. Das et al. [140] highlighted the significant influence of pyrolysis temperature and feedstock type on biochar composition, emphasizing the variability in its properties. Similar to soil taxonomy, standardized protocols and classifications for biochar production can help categorize biochars based on their intended use and predictable outcomes. Collaborative databases that incorporate global research can also assist in understanding this variability.

**Economic viability:** Biochar's high production and transportation costs outweigh its uncertain agricultural benefits, making it a challenge, especially for farmers in developing nations [132,141]. Economies of scale, integrated biochar production within waste management or energy generation systems, and government subsidies can make biochar affordable. The development of low-tech, locally adapted production methods can also be made more accessible to smallholder farmers.

**Potential environmental risks:** Improper production or application of biochar might result in toxin leaching, alteration of soil pH in a detrimental way, or even harm to aquatic ecosystems. This could be due to the release of harmful components or negative interactions with the environment [142,143]. Rigorous quality checks, biochar production and

application guidelines, and continuous monitoring can mitigate these issues. Therefore, educating farmers about the best practices for biochar application is essential.

**Limited knowledge transfer:** Often, there is a gap between the research findings and their applications by end users. This can lead to suboptimal or misguided biochar use, particularly among farmers with limited education or resources [126]. Bridging the gap between researchers, extension services, and farmers through workshops, field days, and accessible literature can ensure the widespread dissemination of the latest findings and best practices.

**Sociocultural acceptance:** Introducing new agricultural practices, such as biochar application, can sometimes lead to resistance due to traditional farming methods or a lack of awareness about its benefits in some regions [126]. Participatory research involving farmers in the research process can increase acceptance. Therefore, integrating cultural and social considerations into biochar promotion strategies is crucial.

Despite these challenges, the benefits of biochar and the dedication of the global research community have shone through. As research continues, tailored solutions are emerging, making the integration of biochar into global agriculture more feasible.

## 9. Conclusions

The potential of biochar for reshaping sustainable agriculture is extensively highlighted in this review. P, a crucial element in agriculture, faces the challenge of fixation in soils with significant environmental and economic repercussions, such as increased greenhouse gas emissions and mounting costs for farmers. Biochar is a promising remedy because of its capacity to adsorb P and minimize its fixation in soils. However, the diversity of biochar properties resulting from different feedstocks and production techniques emphasizes the need for an intricate understanding of uniform results. Progress in biochar modification methods has extended its possibilities, perfecting its attributes for P management. Breakthroughs in analytical tools combining cutting-edge spectroscopy and artificial intelligence have provided comprehensive insights into the interactions of biochar with soil and P. These findings suggest the broad application of biochar derived from P management to enhance soil health and carbon sequestration. By improving soil P availability, crop yields can be increased, and excessive fertilization reduced. This economically benefits farmers and safeguards aquatic ecosystems from P runoff. However, harnessing the full potential of biochar is challenging because of its diverse production and acceptance in traditional agricultural settings. This emphasizes the need for continued research, especially in customizing biochar properties and assessing their long-term implications in diverse soils. The adaptability of modified biochar stands out during this process. Through modification, its characteristics can be tailored to address specific agricultural challenges, making it a potent tool for sustainable farming. However, as we have modified it, caution is necessary to ensure that the environmental integrity of the biochar remains intact. Biochar, especially its modified form, represents hope for a sustainable agricultural future. The insights from this review reinforce the urgency of ongoing research and collaboration, paving the way for a balanced and bountiful agricultural landscape.

**Author Contributions:** The review was conceptually designed with contributions from all authors. N.A. and L.D. (Lifang Deng) jointly led the literature search and data analysis, contributing equally to the work. Z.H.S., along with providing critical revisions, significantly shaped the intellectual content of the review. B.B. and C.W. were essential for evaluating the relevance of the studies included in the review. L.D. (Lansheng Deng), Y.L., J.L. and S.C. were instrumental in interpreting the research findings and providing key revisions to the manuscript. Z.C., F.H., F.A., R.A. and L.G. contributed to the initial drafting and subsequent critical revisions of the manuscript. P.T. coordinated the review process and provided overall guidance, contributing to the critical review of the manuscript for important intellectual content. All authors engaged in the final drafting of the manuscript, provided comments and insights at various stages, and approved the final version of the manuscript. All authors have read and agreed to the published version of the manuscript.

**Funding:** We gratefully acknowledge the financial support from the National Natural Science Foundation of China (No. 42377211) and the Basic and Applied Basic Research Foundation of Guangdong Province (No. 2022A1515010941), Scarce and Quality Economic Forest Engineering Technology Research Center (2022GCZX002), Science and Technology Plan projects of Guangzhou (NO. 202206010069), Meizhou Science and Technology Project (NO. 2021A0304001), and the Key-Area Research and Development Program of Guangdong Province (2020B020215003).

**Data Availability Statement:** No new data were created or analyzed in this study. Data sharing is not applicable in this study.

**Acknowledgments:** During the preparation of this work, the authors used Paperpal in order to check grammar, sentence structure, and punctuation. After using this tool/service, the authors reviewed and edited the content as needed, and take full responsibility for the content of the publication.

**Conflicts of Interest:** Author Lin Gong was employed by the company Dongguan Yixiang Liquid Fertilizer Co., Ltd. The authors declare that they have no affiliations with or involvement in any organization or entity with any financial interest in the subject matter or materials discussed in this manuscript.

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
