# Peer review of "Advancements in Biochar Modification for Enhanced Phosphorus Utilization in Agriculture"

_land, doi:10.3390/land13050644_

Round 1

Reviewer 1 Report

Comments and Suggestions for Authors

This study summarized the advancements in biochar modification for enhanced phosphorus utilization in agriculture.  After carefully reading, the topic of this research is of significance as modified biochar enhances phosphorus (P) availability and is an environmentally friendly approach to address soil P deficiency. The structure of the manuscript is basically logical. However, only a macro qualitative analyses on the P fication challenges and impacts, role of biochar in P adsorption, biochar modification and characterization,  and the role of biochar in sustainable agriculture were performed in the whole text. The specific quantitative analyses (e.g., Meta analysis) on the  role of modified biochar in enhancing phosphorus (P) availability were not found, which greatly reduced the quality of the review manuscript. The authors are encouraged to carry out an in-depth quantitative analyses and review on the advancements in biochar modification for enhanced agricultural phosphorus utilization.

Author Response

Response to Reviewer 1

Dear Reviewer

Thank you for your constructive feedback. We appreciate your recognition of the importance of our topic on advancements in biochar modification for enhanced phosphorus utilization. In response to your comments, we have updated our manuscript to include specific quantitative data regarding phosphorus adsorption by modified biochar:

We revised Table 1 to detail the phosphorus adsorption capacities of modified biochars, including specific values for P adsorption measured under various experimental conditions. This provides a direct quantitative trait of modified biochars, illustrating the enhancements due to biochar modification.

These enhancements aim to strengthen the manuscript by providing the quantitative depth requested and ensuring that the results are contextualized appropriately within the current research landscape.

We believe these revisions address your concerns and enhance the manuscript’s contribution to the field. We are committed to further exploring these quantitative aspects in future research and appreciate the guidance your feedback has provided.

Thank you for considering our revised manuscript.

Best regards,

Reviewer 2 Report

Comments and Suggestions for Authors

This paper reviews the effect of biochar on phosphorus. I think there are still some problems to be improved and optimized.

1. Why is element P studied and what is its particularity in the ecosystem besides its nutritional function?

2. What are the sources and preparations of biochar, etc., can be properly introduced.

3. Please analyze why biochar can improve the activity of P.

4. The authors can appropriately summarize the research progress and existing problems of the effect of biochar on P element.

5. The authors are requested to state the existing problems as far as possible in the process of literature review.

6. In writing, it is recommended to be as concise as possible, to elaborate on key issues, and not to over-analyze basic knowledge.

7. Please summarize the scientific problems worth studying in the future for others to study.

Author Response

Response to Reviewer 2

Dear Reviewer

Thank you for your insightful comments and suggestions. We have carefully considered each point and made corresponding revisions to the manuscript. Below, we summarize our responses to your queries and suggestions:

  1. Why is element P studied and what is its particularity in the ecosystem besides its nutritional function?

Response: We have expanded the discussion on phosphorus’s role beyond plant nutrition, detailing its critical impact on soil health and the broader ecosystem, particularly its contribution to eutrophication and biodiversity loss, as noted in lines 173-191 and 703-712.

  1. What are the sources and preparations of biochar, etc., can be properly introduced.

Response: We clarified the diversity of biochar feedstocks and the conditions under which biochar is prepared, highlighting how these factors influence its phosphorus adsorption capabilities, as detailed in lines 271-299.

  1. Please analyze why biochar can improve the activity of P.

Response: Our manuscript outlines the physicochemical mechanisms through which biochar enhances phosphorus adsorption and availability, including its impact on soil pH and nutrient retention, discussed in lines 192-228.

  1. The authors can appropriately summarize the research progress and existing problems of the effect of biochar on P element.

Response: We provided a comprehensive summary of the advancements and existing challenges in using biochar for phosphorus management, emphasizing the need for more research on its long-term impacts and the variability in biochar’s efficacy due to production conditions.

  1. The authors are requested to state the existing problems as far as possible in the process of literature review.

Response: We identified key challenges in the current literature, such as inconsistencies in methodologies and a lack of long-term studies, which complicate understanding biochar’s sustained effects on phosphorus management.

  1. In writing, it is recommended to be as concise as possible, to elaborate on key issues, and not to over-analyze basic knowledge.

Response: We have revised the manuscript to enhance conciseness and focus, streamlining content and emphasizing key issues without over-analyzing basic knowledge. These changes have improved the manuscript’s readability and directness.

  1. Please summarize the scientific problems worth studying in the future for others to study.

Response: We outlined several promising research areas, including the integration of biochar with sustainable agricultural practices and cutting-edge technologies, and the exploration of biochar in circular economy models. These potential studies are expected to further our understanding of biochar’s benefits and applications in agriculture.

We appreciate your thorough review and believe that these revisions have substantially strengthened the manuscript. We hope that our responses adequately address your concerns and contribute to the clarity and depth of the research presented.

Thank you for your valuable feedback and consideration.

Best regards,

Reviewer 3 Report

Comments and Suggestions for Authors

The paper „Advancements in Biochar Modification for Enhanced Phosphorus Utilization in Agriculture” is current.

I propose the publish of the paper after some minor correction like:

-please put at the end of the first chapter a sentence referring to the purpose of the article.

Author Response

Response to Reviewer 3

Dear Reviewer

Thank you for your positive evaluation of our paper "Advancements in Biochar Modification for Enhanced Phosphorus Utilization in Agriculture" and for your suggestion to clarify the purpose of the article at the end of the first chapter.

We agree that explicitly stating the purpose can enhance reader understanding and provide clear direction for the content that follows. To address your suggestion, we have added the following sentence to the end of the first chapter:

"The purpose of this article is to comprehensively review recent advancements in biochar modifications aimed at enhancing phosphorus utilization in agriculture, identify the existing gaps in the research, and suggest directions for future studies."

We believe this addition effectively sets the stage for the discussions that follow and clearly outlines the scope of the article, aligning with its objectives.

Thank you again for your constructive feedback, which has helped improve the clarity and impact of our manuscript.

Best regards,

Reviewer 4 Report

Comments and Suggestions for Authors

The paper reviewed the knowledge in the field of biochar modifications for a better utilization of phosphorus in agriculture as an alternative of using P feritilizers

The paper reviewed the knowledge in the field of biochar amendments for a better use of phosphorus in agriculture as an alternative to the use of traditional phosphate fertilizers which are expensive and present environmental hazards due to P fixation and leaching.

The review was a complex work and systematized information of 140 papers. The gaps of knowledge were highlighted, and future directions of research were proposed.

The methods of reviewing the literature ( key words used for search, the time interval of the of the revised publications  should be presented in the paper).

A recent paper that treated reviewed the biochar -based slow-release of fertilizers was not cited:

https://doi.org/10.1016/j.ese.2022.100167 2666-4984

Minor observation:

1)Page 1 , lines 22-24

The phrase “By adjusting properties such as pH  levels and functional groups to align with the phosphate zero point of charge, the ability of biochar to adsorb and retain P, increasing its bioavailability to plants.” should be revised  and corrected .

Author Response

Response to Reviewer 4

Dear Reviewer

Thank you for your thorough review and constructive comments on our manuscript titled "Advancements in Biochar Modification for Enhanced Phosphorus Utilization in Agriculture." We appreciate your recognition of the systematic and comprehensive approach of our review, which analyzed 143 papers. We have carefully considered your suggestions and have made the following revisions to our manuscript:

  1. Upon your recommendation, we have reviewed and incorporated the recent paper by Wang et al. (2022) on biochar-based slow-release fertilizers, which is indeed highly relevant to our discussion. This citation has been added to two sections of our manuscript to enrich the discourse on biochar's mechanisms and future research directions:
    • Section 3.1: Basic Mechanism of P Adsorption by Biochar
    • Section 5.2: Functional Groups in Biochar

The citation details are as follows:

Wang, C.; Luo, D.; Zhang, X.; Huang, R.; Cao, Y.; Liu, G.; Zhang, Y.; Wang, H. Biochar-Based Slow-Release of Fertilizers for Sustainable Agriculture: A Mini Review. Environmental Science and Ecotechnology 2022, 10, 100167, doi:10.1016/J.ESE.2022.100167.

  1. We have addressed your observation concerning the clarity and grammatical structure of the sentence on Page 1, lines 22-24. The sentence has been revised to: "By adjusting properties such as pH levels and functional groups to align with the phosphate's zero point of charge, we enhance biochar's ability to adsorb and retain phosphorus, thereby increasing its bioavailability to plants."

We hope these revisions adequately address your concerns and contribute further to the strength and clarity of our manuscript. Thank you again for your valuable feedback, which has undoubtedly improved the quality of our work.

Best regards,

Round 2

Reviewer 1 Report

Comments and Suggestions for Authors

All comments and suggestions have been addressed in the revised version of the manuscript. The manuscript is currently significantly improved and can be considered to be accepted for publication.